# Gender-based heterogeneity of FAHFAs in trained runners

**Alisa B. Nelson**[1], **Lisa S. Chow**[2], **Donald R. Dengel**[3], **Meixia Pan**[4,5], **Curtis C. Hughey**[1], **Xianlin Han**[4,5], **Patrycja Puchalska**[1], **Peter A. Crawford**[1,6]*

**1** Division of Molecular Medicine, Department of Medicine, University of Minnesota, Minneapolis, MN, United States of America, **2** Division of Diabetes, Endocrinology and Metabolism, Department of Medicine, University of Minnesota, Minneapolis, MN, United States of America, **3** School of Kinesiology, University of Minnesota, Minneapolis, MN, United States of America, **4** Barshop Institute for Longevity and Aging Studies, University of Texas Health Science Center at San Antonio, San Antonio, TX, United States of America, **5** Department of Medicine—Diabetes, University of Texas Health Science Center at San Antonio, San Antonio, TX, United States of America, **6** Department of Biochemistry, Molecular Biology, and Biophysics, University of Minnesota, Minneapolis, MN, United States of America

* crawforp@umn.edu

**Data Availability Statement:** All of the data used for this independent and fully separate analysis are already publicly available through: https://insight.jci.org/articles/view/158037 (PMID: 35192550,

## Abstract

Fatty acid esters of hydroxy fatty acid (FAHFA) are anti-diabetic and anti-inflammatory lipokines. Recently FAHFAs were also found to predict cardiorespiratory fitness in a cross-sectional study of recreationally trained runners. Here we report the influences of body composition and gender on static FAHFA abundances in circulation. We compared the association between circulating FAHFA concentrations and body composition, determined by dual x-ray absorptiometry, in female recreational runners who were lean (BMI < 25 kg/m², n = 6), to those who were overweight (BMI ≥ 25 kg/m², n = 7). To characterize the effect of gender we also compared circulating FAHFAs in lean male recreational runners (n = 8) to recreationally trained lean female (n = 6) runner group. Circulating FAHFAs were increased in females in a manner that was modulated by specific adipose depot sizes, blood glucose, and lean body mass. As expected, circulating FAHFAs were diminished in the overweight group, but strikingly, within the lean cohort, increases in circulating FAHFAs were promoted by increased fat mass, relative to lean mass, while the overweight group showed a significantly attenuated relationship. These studies suggest multimodal regulation of circulating FAHFAs and raise hypotheses to test endogenous FAHFA dynamic sources and sinks in health and disease, which will be essential for therapeutic target development. Baseline circulating FAHFA concentrations could signal sub-clinical metabolic dysfunction in metabolically healthy obesity.

## Introduction

Fatty acid esters of hydroxy fatty acids (FAHFAs) form a lipid class in which each species is composed of a fatty acyl chain esterified to a hydroxy fatty acid. These lipids serve as signaling lipokines with insulin-sensitizing and anti-inflammatory effects [1–5]. While FAHFA

 

PMCID: PMC9057596) AND in S1 Table provided herein.

**Funding:** NIH DK091538, AG069781, DK098203, T32DK007203, P30 AG013319, P30 AG044271. The funders had no role in study design, data collection and analysis, decision to publish, or preparation of the manuscript. Thank you for changing the online submission form on our behalf.

**Competing interests:** P.A.C. has served as an external consultant for Pfizer, Inc., Abbott Laboratories, Janssen Research & Development and Selah Therapeutics. I confirm that these competing interest do not alter our adherence to PLOS ONE policy on sharing data and material.

concentrations vary among rodent tissues and tend to be more abundant in humans with lower body mass index (BMI) and higher insulin sensitivity, their regulation and mechanisms of action are not yet fully understood [1, 6–8]. We previously measured the concentrations of 25 FAHFA species using targeted multidimensional mass spectrometry-based shotgun lipidomics in a cohort of recreationally trained runners across a range of BMI to investigate signatures that might identify metabolically healthy obese (MHO) participants [9]. As expected, baseline circulating FAHFAs (participants at rest and after ≥8-hour fast) were negatively associated with BMI and total fat mass. FAHFAs were dynamically regulated during acute aerobic exercise in lean participants, but they were largely unchanged in overweight or obese participants, indicating a concealed effect of obesity in an otherwise MHO group [10]. We hypothesized that circulating FAHFAs reflect differences in body composition. Therefore, in this study, we investigated relationships between baseline circulating FAHFAs in recreationally trained runners relative to dual x-ray absorptiometry (DXA) based body composition measurements.

## Methods and materials

### Participant details

Overweight recreationally trained (OWT; n = 11) and normal weight recreationally trained (NWT; n = 14) participants who self-reported aerobic exercise (3–5 sessions/week) from the Twin Cities metro area were recruited between July 2014 and April 2017. We preferentially recruited participants from recent running events, to ensure that they could complete a prolonged (90 minute) run. Inclusion criteria were: 1) Age 18–40 years, and 2) Regular aerobic exercise, preferably running, at least 3–5 sessions/week. Individuals with 1) Self-reported clinically significant medical issues (for example diabetes, cardiovascular disease, uncontrolled pulmonary disease), 2) abnormal EKG indicating cardiac disease (study EKG performed) and 3) current pregnancy (screening pregnancy test performed) were excluded. The average VO2max of NWT and OWT participants was measured as previously published [9]. In the OWT group, it was 62.0±2.1 (mL/kg lean mass/min), whereas in the NWT group, it was 73.3±5.3 (mL/kg lean mass/min). Participants were recruited with the goal to achieve similarity in age and sex between the two groups.

For baseline blood sample collection, participants were instructed to avoid intentional exercise for two days and arrived for the visit after an overnight fast (minimum 8 hours). These samples were used to measure baseline FAHFA concentrations as well as insulin and glucose levels. Insulin sensitivity was estimated by homeostatic model assessment for insulin resistance [HOMA-IR: (fasting serum insulin (uU/mL) fasting glucose (mmol/L))/22.5] [11, 12]. Division into gender-based subgroups were based on participant self-identification and BMI. Of the 11 participants in the OWT group, 1 male participant showed lower body fat via DXA scan, suggesting an elevated weight-to-height ratio due to increased lean body mass, not fat mass. Due to the associative nature of these analyses, this participant was not excluded. The University of Minnesota's Institutional Review Board (IRB) approved the study protocol and methods. The study was conducted in accordance with the Declaration of Helsinki. All participants provided written informed consent before study participation. Participant data was de-identified for all data analysis.

### Dual X-ray absorptiometry

Lean mass, bone mass, bone mineral content, and fat mass were measured by dual X-ray absorptiometry (DXA) using a GE Healthcare Lunar iDXA (GE Healthcare Lunar, Madison, WI) with Encore software (version 16.2). The DXA scan was performed in the fasted state

 

during a separate visit. Regional measures of trunk, android, gynoid, abdominal subcutaneous adipose tissue (SAT), and visceral adipose tissue (VAT) were made. Trunk fat mass included fat mass from the chest, abdomen, and pelvis region. The android region was defined as the trunk area approximately between the ribs and the pelvis. The upper boundary was set at 20% of the distance between the iliac crest and the base of the skull. The lower boundary was the top of the iliac crest. The gynoid region included the hips and upper thighs, overlapping both the leg and trunk regions. SAT in the android region was determined by examining the X-ray attenuation between the edge of the body and the outer edge of the abdominal cavity [13]. VAT in the android region was calculated by subtracting android SAT from android total fat mass, as previously described [13].

## Quantitation of FAHFA

Quantitation of FAHFA was previously described [9, 14]. Briefly, FAHFA were identified and quantified in serum through multidimensional MS-based targeted shotgun lipidomics using appropriate amount of internal standard 12-PAHSA-d4 (**S1 Table**). Lipids were extracted using a modified Bligh and Dyer protocol and solid phase extraction with a HyperSep silica SPE cartridge at room temperature. Before infusion, FAHFAs were derivatized with N-[4-(Aminomethyl)phenyl]pyridinium (AMPP). Then, derivatized extracts were infused in TSQ triple quadrupole mass spectrometer (Thermo Fisher Scientific) equipped with an automated nanospray device (Triverse Nanomate, Advion Biosciences, Ithaca, NY). Identification and quantification were determined by product-ion analysis. Optimized collision-induced dissociation was also used for neutral loss scanning.

## Statistics

FAHFA concentration was auto scaled. Briefly, the average concentration was computed for each FAHFA species based on all 25 participants. Each sample value was then mean-centered and divided by the standard deviation of each variable. Group differences in body composition, and FLR analyzed using Ordinary one-way ANOVA with Tukey's multiple comparisons test. Correlation analysis between raw concentrations of individual FAHFA species and body composition were computed using the Pearson correlation in GraphPad Prism v. 9.5.1 for (i) all female participants (n = 13); (ii) all NWT participants (n = 14); (iii) all OWT participants (n = 11) and corrected for multiple comparisons using Benjamini-Hochberg method. Prediction of total circulating FAHFAs by simple or multiple regression (~) of continuous independent variables BMI or LBM and fat mass was performed on standardized data with the inclusion of interaction terms (BMI:Gender, LBM:FM). Gender was included as a binary independent variable, 1 for female and 0 for male. For ANCOVA, the tested outcome was total circulating FAHFAs, investigating the main effect of BMI as a binary independent variable with the covariate LBM or gynoid mass as continuous independent variables. All male and female participant data were combined by BMI $< 25$ kg/m$^2$ or BMI $\geq 25$ kg/m$^2$. A significant difference in slope and y-intercept by BMI group after adjusting the average circulating total FAHFA abundance for covariates LBM or gynoid mass was determined by F statistic.

## Results and discussion

### Baseline circulating levels of FAHFAs in trained runners are associated with gender, but only in lean individuals

To determine the relationships between body composition and circulating FAHFA concentrations in lean and overweight recreationally trained runners, we first performed a linear

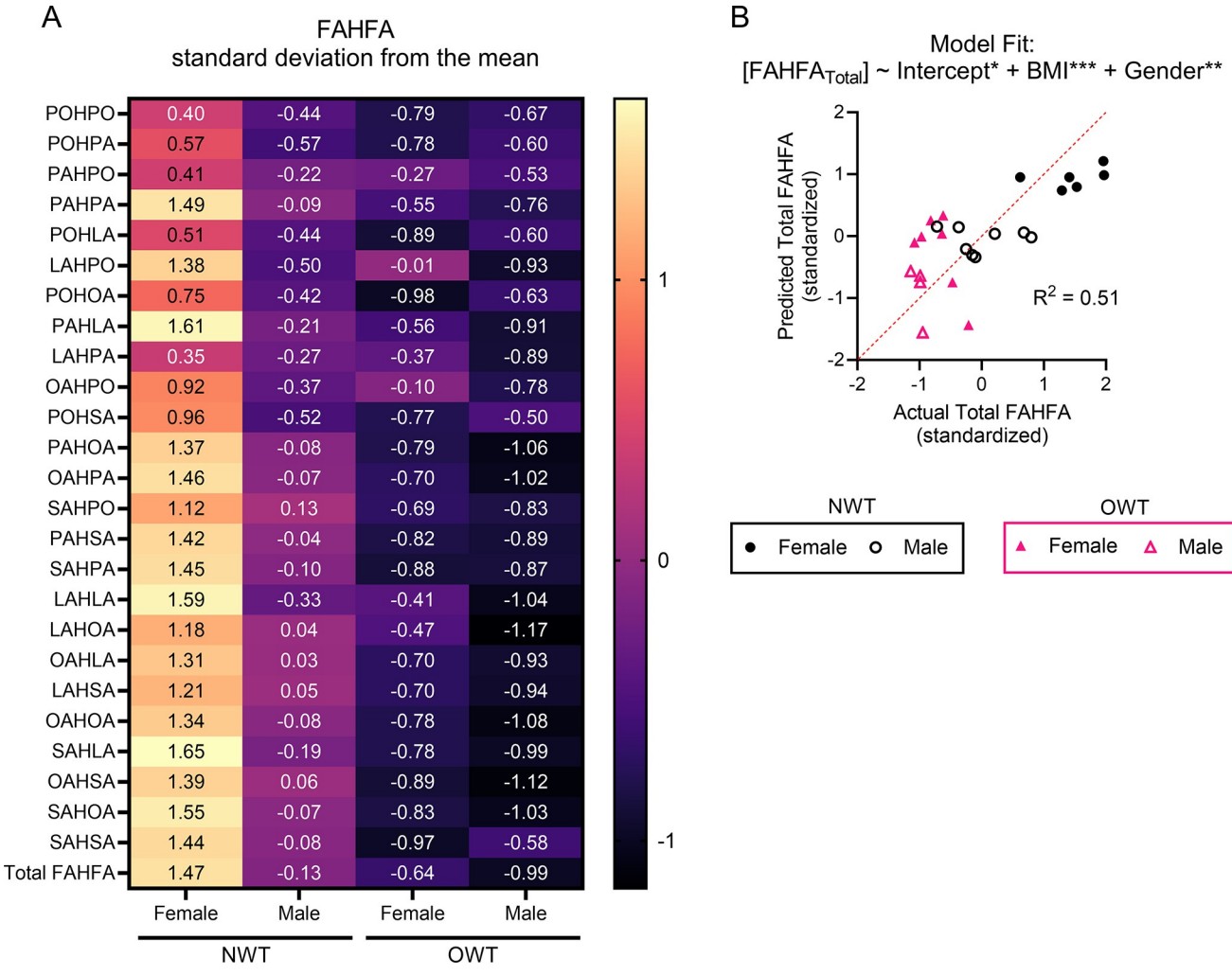

**Fig 1. Gender differences in baseline circulating FAHFAs in lean, but not overweight, runners. (A)** Autoscaled (mean-centered, divided by standard deviation) of circulating concentration of FAHFA species of female and male participants of normal weight-trained (NWT) and overweight and obese trained (OWT) groups. The numerical label indicates deviation from the mean of FAHFA concentration across all participants for each subgroup. **(B)** Actual versus predicted total circulating FAHFAs for linear regression model: Total FAHFA modeled by (~) independent variables BMI (continuous) and Gender (binary: female = 1, male = 0), where the intercept indicates average total FAHFA in a male participant with average BMI. *$p<0.05$, **$p<0.01$, ***$p<0.001$, ****$p<0.0001$ using One-way ANOVA.

regression of BMI alone to predict FAHFAs (Total FAHFAs = $\beta_0$ + $\beta_1$*BMI; $\beta_1$ = -0.55, 95% CI [-0.91 to -0.19], p-value = 0.0044), and found that BMI alone could moderately predict total circulating FAHFAs ($R^2$ = 0.3) (goodness of fit shown in **S1A Fig**). Variability of circulating FAHFAs within the NWT group led us to investigate whether a gender-based (participant self-identified) distinction in circulating FAHFAs could be observed. After pooling all male, female, normal weight recreationally trained (NWT, BMI < 25 kg/m$^2$), and overweight or obese recreationally trained (OWT, BMI $\geq$ 25 kg/m$^2$) participant data per FAHFA species, serum concentrations of many FAHFA species were significantly higher than the mean in female NWT runners compared to male NWT (**Fig 1A**). Meanwhile, the OWT group did not show any differences in circulating FAHFAs between male and female participants, indicating that gender alone does not account for FAHFA abundance. These results suggest the possibility of both BMI and gender effects on circulating FAHFAs. Multiple linear regression showed

the effect of BMI adjusted for gender significantly improved model fit (Total FAHFA = $\beta_0$ + $\beta_1$*BMI + $\beta_2$*Gender, $R^2$ = 0.51; BMI: $\beta_1$ = -0.65, 95% CI: [-0.97 to -0.33], p-value = 0.0003; Gender: $\beta_2$ = 0.91, 95% CI: [0.29 to 1.53], p-value 0.006) (goodness of fit shown in **Fig 1B**). Inclusion of an interaction variable (denoted by BMI: Gender; Total FAHFA = $\beta_0$ + $\beta_1$*BMI + $\beta_2$*Gender + $\beta_3$*BMI:Gender) was not significant (goodness of fit shown in **S1B Fig**). This model suggests that recreationally trained female participants have elevated FAHFAs in circulation compared to males, which are attenuated for each unit increase in BMI.

## FAHFAs are differentially associated with body composition

Although FAHFA concentrations have been quantified in numerous rodent tissues, including various adipose depots, liver, and skeletal muscle, little is known about their turnover (*i.e.*, specific sites for FAHFA production and consumption) [4, 8]. Studies investigating enzymes involved in their production suggest FAHFAs originate from adipose tissue, with other tissues acting as sites of disposal, but this has yet to be elucidated in physiologically dynamic states [1, 3, 6, 7, 15]. Association analyses may lead to hypotheses of FAHFA dynamics, possibly conferring the roles of source and sink to adipose and lean tissues, respectively. Both BMI and gender are known to influence adiposity and lean mass (i.e., distribution and size) [16]. As we observed both gender and BMI-related differences in circulating FAHFAs, we hypothesized adipose depots may underlie these differences (**Table 1**). While the OWT group had greater adiposity than the NWT group, OWT females had significantly more total fat mass than OWT males (**Fig 2A**). OWT females also had greater gynoid fat mass than all other participants (**Fig 2B**). Visceral adipose tissue (VAT) of the android compartment, is associated with cardiometabolic disease risk [17]. Female participants of the NWT group had the lowest VAT (78.8 ± 28.5g). OWT male participants (740.8 ± 178.3g) had 1.4-fold greater VAT than OWT females (524.0 ± 61.7g), 3.4-fold greater than NWT males (217.1 ± 35.0g), and 9-fold greater compared to NWT females (**Fig 2C**). Finally, male participants in both NWT and OWT had significantly more lean body mass (LBM) than corresponding females (**Fig 2D**).

To investigate the relationship between body composition and circulating FAHFA pool sizes we sought to isolate these effects. We employed the Pearson correlation method between fat mass depots and LBM with circulating FAHFAs in (i) female (all NWT + OWT) participants (rationale: females were more dynamic comparing NWT to OWT groups for the effect of BMI, **Fig 3A**) and (ii) all NWT (male + female) participants (rationale: the gender difference was more pronounced in the NWT group, **Fig 3B**).

**Table 1. Participant characteristics.**

| Mean (SE) | NWT (N = 14) | | OWT (N = 11) | |
|---|---|---|---|---|
| | Male (N = 8) | Female (N = 6) | Male (N = 4) | Female (N = 7) |
| Age (years) | 30.2 (2.0) | 26.6 (2.5) | 30.3 (3.1) | 33.0 (1.8) |
| BMI (kg/m$^2$) | 22.2 (0.6) | 21.5 (0.6) | 29.1 (2.0)[a,b] | 31.5 (2.0)[a,b] |
| Lean body mass (kg) | 60.5 (2.1) | 45.4 (1.8)[a] | 76.1 (3.4)[a,b] | 54.7 (3.3)[c] |
| Total fat mass (kg) | 9.6 (0.9) | 14.8 (4.0) | 21.6 (4.3)[b] | 36.0 (3.4)[a,b,c] |
| Android fat (g) | 569.5 (111.8) | 772.0 (117.9) | 2033 (593.7)[a] | 2853 (371.7)[a,b] |
| VAT (g) | 217.1 (35.0) | 78.8 (28.5) | 740.8 (178.3)[a,b] | 524.0 (61.7)[a,b] |
| SAT (g) | 352.6 (86.9) | 693.2 (114.4) | 1293 (415.9) | 2329 (357.1)[a,b] |
| Gynoid (g) | 1386 (159.7) | 3040 (353.7) | 3425 (586.4) | 6967 (765.7)[a,b,c] |
| HOMA-IR | 0.86 (0.13) | 0.98 (0.25) | 1.97 (0.27)[a] | 1.35 (0.30) |

a: *p-value < 0.05 compared to NWT males;* b: *p-value < 0.05 compared to NWT females;* c: *p-value < 0.05 compared to OWT males.*

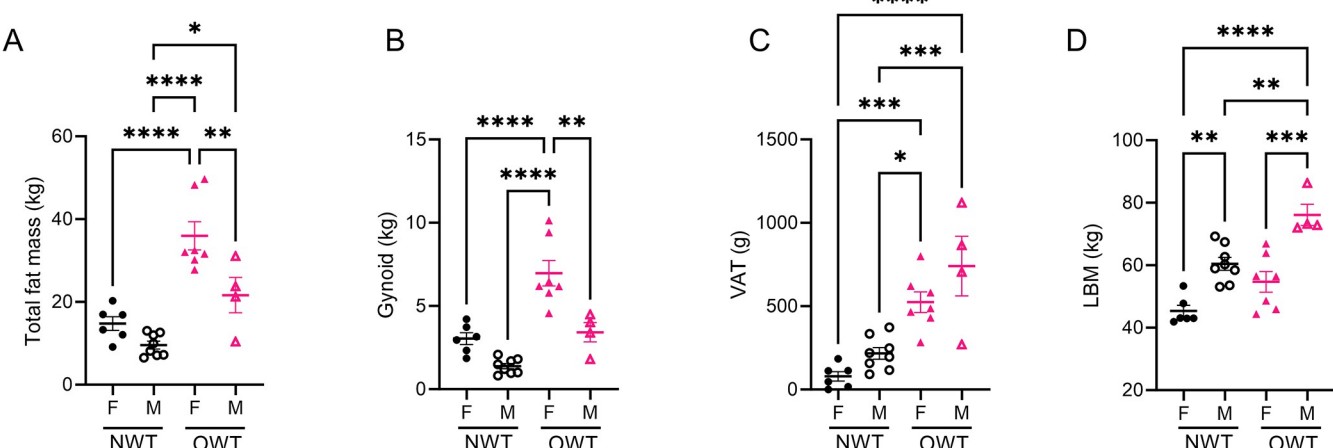

**Fig 2. Gender differences in body composition of recreationally trained runners. (A)** Total fat mass; **(B)** gynoid fat mass; **(C)** visceral adipose tissue (VAT) fat mass; **(D)** lean body mass (LBM) measured by DXA. *$p<0.05$, **$p<0.01$, ***$p<0.001$, ****$p<0.0001$ using One-way ANOVA with Tukey's multiple comparisons test. Error bars represent SEM.

When examining all (NWT + OWT) female participants (n = 13) many FAHFA species showed strong negative associations with all fat mass depots (red box indicates those with adj. $p < 0.05$ after correcting for multiple comparisons) **(Fig 3A)**. Significantly strong negative relationships between 18 out of 25 FAHFAs [including the most widely studied FAHFA, palmitic acid ester of hydroxy stearic acid, PAHSA, (R = -0.69 [95% CI: -0.90 to -0.23], $p = 0.009$)] and total fat mass were observed. No significant differences in the strength of these relationships were seen among specific adipose tissue depots. These findings indicate that in this cohort of recreationally trained females, higher fat mass was associated with lower circulating FAHFAs in a manner that does not appear to be associated with a specific depot. This result is underscored by a significantly negative association with percent body fat and a positive association with percent lean mass, which is in agreement with our previous analysis of the full 25-participant data set [9]. Interestingly, five FAHFA species (PAHPO, POHLA, POHOA, LAHPA, and POHSA, but not PAHSA) negatively correlated with circulating insulin, glucose, or HOMA-IR values **(Fig 3A)**. LAHPA showed the strongest negative relationship with fasting glucose (R = -0.78 [95% CI: -0.93 to -0.40], $p = 0.002$). However, none of these five species was negatively associated with fat mass in the lean, trained female participants (LAHPA with Total fat mass: R = -0.38 [95% CI: -0.77 to 0.22], $p = 0.2$) [11, 12]. This unexpected divergence suggests these species might be independently regulated by hormone-fuel relationships rather than adiposity. This analysis was repeated in all male participants and showed similar associations with body fat composition, indicating increased body fat in proportion to lean mass has a negative relationship with circulating FAHFAs **(S2A Fig)**.

Strikingly, correlation analysis of the isolated NWT (male + female) group (n = 14) shows that 14 out of 25 FAHFA species (including PAHSA) had negative associations with LBM and percent lean mass (Pearson correlation coefficient, R < -0.5; white box indicates those with adj. $p < 0.1$ after correcting for multiple comparisons) **(Fig 3B)**. Given prior positive associations of circulating FAHFAs with leanness, this observation, generated in a focused analysis of lean, recreationally trained human participants, was unexpected [9, 18]. Moreover, while VAT retained its negative association to FAHFAs, relationships to total fat mass, SAT fat (13 of 25 species, R ≥ 0.5), and gynoid fat (13 of 25 species R ≥ 0.5) were positively associated with baseline circulating FAHFA concentrations in the NWT cohort, which was also unexpected.

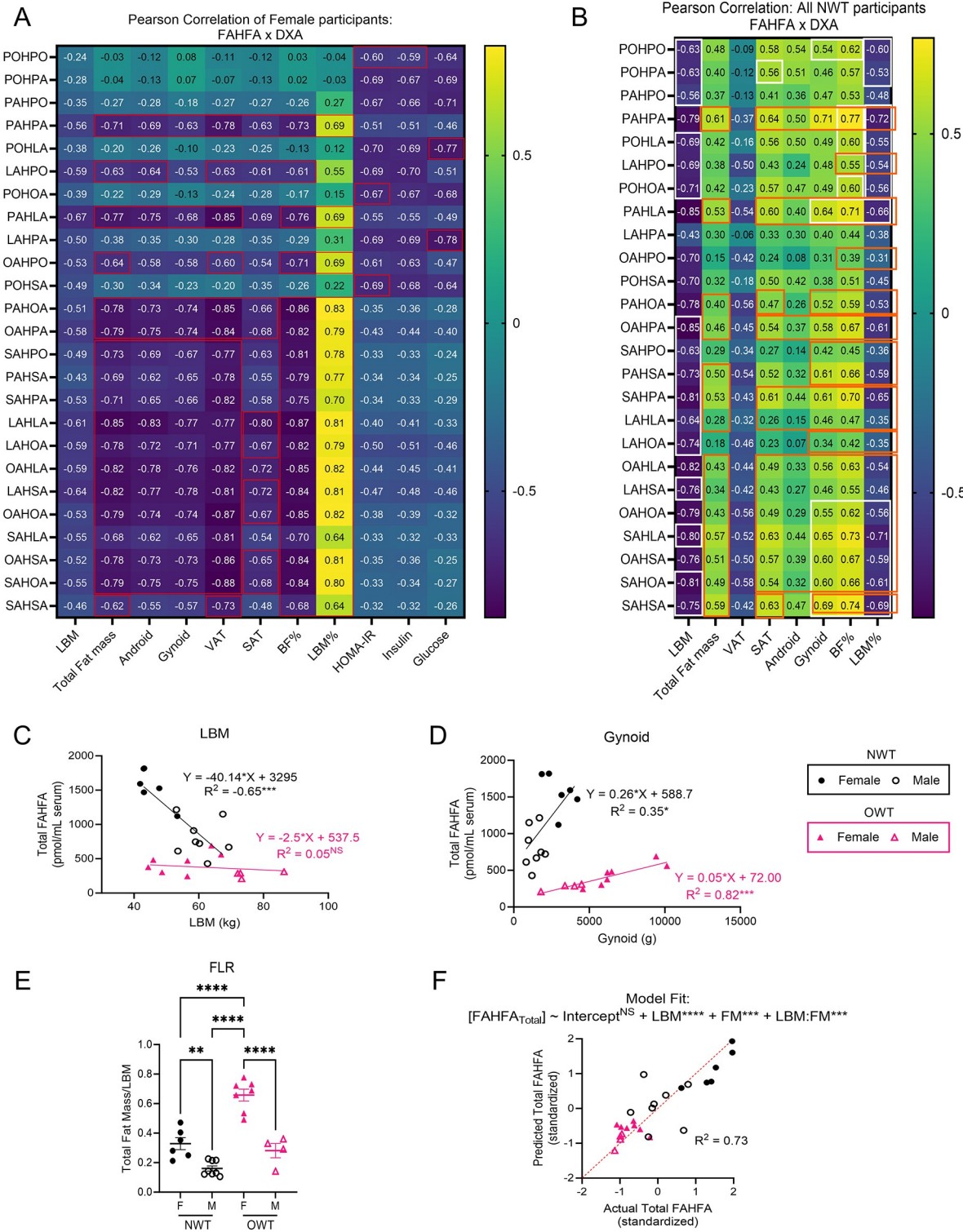

**Fig 3. Differential associations between FAHFAs and body composition suggest multimodal influence of FAHFAs by gender and BMI. (A)** Pearson correlation coefficients of raw circulating concentrations of FAHFA species with DXA measurements, calculated HOMA-IR score, circulating insulin, and circulating glucose for all (NWT + OWT) female participants (n = 13). **(B)** Pearson correlation coefficients of raw circulating concentrations of FAHFA species with DXA measurements for male and female participants (NWT group only, n = 14). |R > 0.5| have a $p < 0.05$; Red box: adj. $p < 0.05$; White box: adj. $p < 0.1$; Orange box: differential correlation between Female and NWT: $p < 0.01$. Scatter plots with fitted linear regression lines of **(C)** LBM and **(D)** gynoid fat mass against total circulating FAHFAs by

BMI group; (**E**) total fat to LBM ratio (FLR). (**F**) Actual versus predicted total circulating FAHFAs after multiple linear regression model: Total FAHFA modeled (~) by multiple independent variables, LBM + FM + their interaction (LBM:FM), where the intercept indicates the average total FAHFA in a participant with average LBM and FM. NS = not significant, *$p<0.05$, **$p<0.01$, ***$p<0.001$ using One-way ANOVA with Tukey's multiple comparisons test. Error bars represent SEM.

We compared Pearson correlation coefficients between Female only (**Fig 3A**) and NWT only (**Fig 3B**) groups to assess the relative impact of BMI and gender, respectively, on the FAHFA-adiposity relationship. Differential associations ($R_{Female}$ in **Fig 3A** vs $R_{NWT}$ in **Fig 3B**, orange boxes indicate $p < 0.01$) between circulating FAHFAs and specific fat depot sizes strongly suggest that excess adiposity alters the regulation of FAHFA turnover (i.e., production and/or consumption) (**Fig 3B**). The interaction between body composition and gender can be further observed by comparing scatterplots of LBM and Gynoid fat mass against total FAHFAs (**Fig 3C, 3D**). We observed distinct clustering of circulating FAHFAs relative to LBM and gynoid fat mass by BMI groups, suggesting distinct relationships between body composition and FAHFAs in the setting of obesity. To test this hypothesis, we fitted regression lines by BMI group and used ANCOVA to determine if the differences slopes and y-intercepts were statistically significant. While the NWT group shows a significant inverse relationship between FAHFAs and LBM ($R^2 = 0.65$, p = 0.0005), the OWT shows no relationship ($R^2 = 0.05$, p = 0.52). After adjusting for the covariate, LBM, the effect of BMI on circulating FAHFAs was significant (p = 0.0002) (**Fig 3C**, **S2B Fig**). The gynoid compartment shows a direct relationship with total circulating FAHFAs in both the NWT and OWT group (**Fig 3D**). Regression shows OWT has a distinct slope and y-intercept from NWT (NWT: $R^2 = 0.35$, p = 0.03); OWT: $R^2 = 0.82$, p = 0.0001) (**Fig 3D**). After adjusting for gynoid mass, BMI showed a significant effect on FAHFA abundance (p < 0.0001). Altogether these results point to a complex and multivariable effect of lean body mass and fat mass on circulating FAHFA species pools that may be driven by gender and differences in body composition.

The distinct clustering of circulating FAHFA distributions by BMI and their relationship to fat mass and lean body mass suggests that excess adiposity corrupts the regulation of FAHFA turnover between source and sink. The anthropometric index, total fat to lean mass ratio (FLR), was previously reported to correlate with cardiometabolic disease [16]. In our study population, FLR is highest in OWT females (**Fig 3E**). Greater FLR in females than males, and higher FAHFA levels in females than BMI-matched males, suggests that increases in circulating FAHFAs are normally promoted by increased fat mass relative to LBM. While this unexpected relationship is evident independently in both lean and overweight groups (**Fig 3D**), the relative impact is significantly diminished in the setting of obesity. Therefore, we hypothesized that a regression model using LBM, total fat mass (FM), and their interaction (LBM: FM) may better predict static FAHFA pools than BMI alone for recreationally trained runners. The resulting model showed a significant contribution of LBM, FM and their interaction (LBM: FM) to predicting total circulating FAHFAs with an $R^2 = 0.73$ (LBM: $\beta_1 = -0.66$, 95% CI: [-0.9 to -0.42], p-value < 0.0001; FM: $\beta_2 = -0.51$, 95% CI: [-0.76 to -0.26], p-value = 0.0004; LBM x FM: $\beta_3 = 0.65$, 95% CI: [0.34 to 0.95], p-value = 0.0002) (model goodness of fit shown in **Fig 3F**). This model suggests that FAHFA abundance in circulation is a function of the relationship between LBM and total fat mass, which is disrupted in obesity. Thus, it is likely that pathogenic adipose expansion, particularly in VAT, which showed an inverse relationship to FAHFAs even in lean participants, negatively influences circulating FAHFA concentrations. These relationships may herald clinical markers of metabolic dysfunction in MHO.

These analyses suggest that circulating FAHFA levels reflect complex regulation influenced by both gender and BMI. Previous work that examined fewer FAHFA species did not identify

gender differences in circulating FAHFAs [18]. Our study involved recreationally trained runners, capable of continuous running for 90 minutes at 60% of their VO$_2$max [9]. Whether these differences are unique to trained humans remains unknown and requires further study to determine how specific compartments dynamically produce and consume FAHFAs in varying states of adiposity and in specific lean and adipose tissue depots. These data also support the notion that select FAHFA lipid species may be independently regulated, and future studies will determine FAHFA lipid species-specific turnover across BMI groups, training status, and gender to parse the multidimensional regulation of circulating FAHFAs. Altogether, these studies raise hypotheses to test endogenous FAHFA dynamic sources and sinks in health and disease, which will be essential for therapeutic target development, and reveal that baseline circulating FAHFA concentrations could signal sub-clinical metabolic dysfunction in MHO.

Limitations of this study include its small sample size, lack of historical data on the training experience and diet of participants, and a lack of an age-matched nontrained cohort. The study is strengthened using a quantitative, targeted lipidomics platform to measure FAHFAs in human serum. Previous work from our lab has shown that untargeted metabolomics analytical methods are susceptible to the formation of fatty acid dimers that mimic FAHFAs, thus confounding interpretation [19]. Here we use both solid phase extraction and careful identification of FAHFAs by selected ion monitoring to ensure proper identification and quantification of this lipid class and their species.

## Supporting information

**S1 Fig.** Actual standardized total FAHFA abundance versus predicted total circulating FAHFAs from regression model (~) (A) univariate effect of BMI, where the intercept indicates average circulating FAHFA for a participant with average BMI, and (B) effect of BMI adjusted for gender and the interaction between BMI: Gender, where intercept represents average FAHFA for male participant with average BMI. $R^2$ value represents goodness of fit. **$p<0.01$, NS: not significant using One-way ANOVA with Tukey's multiple comparisons test.
(TIF)

**S2 Fig.** Heatmaps of Pearson correlation coefficients of FAHFAs with: (A) body composition in male participants; (B) body composition in all OWT participants. Labels indicate Pearson correlation coefficients (R). Species with an $|R > 0.56|$ have an unadjusted $p < 0.05$, however none maintained significance after correction for multiple comparisons.
(TIF)

**S1 Table. FAHFA concentrations in NWT and OWT, male and female participants.** Data presented in pmol / mL of serum and expressed as mean ± standard error of the mean (SEM).
(XLSX)

## Acknowledgments

We acknowledge and appreciate those who participated in our study.

## Author Contributions

**Conceptualization:** Alisa B. Nelson, Patrycja Puchalska, Peter A. Crawford.

**Data curation:** Alisa B. Nelson, Peter A. Crawford.

**Formal analysis:** Alisa B. Nelson, Patrycja Puchalska, Peter A. Crawford.

**Funding acquisition:** Xianlin Han, Peter A. Crawford.

**Investigation:** Alisa B. Nelson, Lisa S. Chow, Meixia Pan, Curtis C. Hughey, Xianlin Han, Patrycja Puchalska, Peter A. Crawford.

**Methodology:** Alisa B. Nelson, Donald R. Dengel, Meixia Pan, Curtis C. Hughey, Xianlin Han, Patrycja Puchalska, Peter A. Crawford.

**Project administration:** Lisa S. Chow, Patrycja Puchalska, Peter A. Crawford.

**Resources:** Lisa S. Chow, Peter A. Crawford.

**Supervision:** Peter A. Crawford.

**Writing – original draft:** Alisa B. Nelson, Lisa S. Chow, Patrycja Puchalska, Peter A. Crawford.

**Writing – review & editing:** Alisa B. Nelson, Lisa S. Chow, Donald R. Dengel, Curtis C. Hughey, Xianlin Han, Patrycja Puchalska, Peter A. Crawford.

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
