## [Decision Letter · Decision Letter 0]

17 Oct 2023

PONE-D-23-18993Gender-based heterogeneity of FAHFAs in trained runnersPLOS ONE

Dear Dr. Crawford,

Thank you for submitting your manuscript to PLOS ONE. After careful consideration, we feel that it has merit but does not fully meet PLOS ONE’s publication criteria as it currently stands.  Two expert Reviewers have raised a number of important points that you should address prior to being re-evaluated.  Particularly, there a revision will require re writing the interpretation of many of the results as this was often confusing. Therefore, we invite you to submit a revised version of the manuscript that addresses the points raised during the review process.

We look forward to receiving your revised manuscript.

Kind regards,

Stephen E Alway, Ph.D.

Academic Editor

PLOS ONE

Journal Requirements:

"NIH DK091538, AG069781, DK098203, T32DK007203, P30 AG013319, P30 AG044271"

"We acknowledge and appreciate those who participated in our study. This work was supported by these funding sources: NIH DK091538, AG069781, DK098203, T32DK007203, P30,AG013319, P30 AG044271"

"NIH DK091538, AG069781, DK098203, T32DK007203, P30 AG013319, P30 AG044271"

6. Thank you for stating the following in the Competing Interests section: 

"P.A.C. has served as an external consultant for Pfizer, Inc., Abbott Laboratories, Janssen Research & Development and Selah Therapeutics."

7. In your Data Availability statement, you have not specified where the minimal data set underlying the results described in your manuscript can be found. PLOS defines a study's minimal data set as the underlying data used to reach the conclusions drawn in the manuscript and any additional data required to replicate the reported study findings in their entirety. All PLOS journals require that the minimal data set be made fully available. For more information about our data policy, please see http://journals.plos.org/plosone/s/data-availability.

8. Please amend the manuscript submission data (via Edit Submission) to include authors Alisa B. Nelson, Lisa S. Chow, Donald R. Dengel, Meixia Pan, Curtis C. Hughey, Xianlin Han and Patrycja Puchalska.

9. Please include your full ethics statement in the ‘Methods’ section of your manuscript file. In your statement, please include the full name of the IRB or ethics committee who approved or waived your study, as well as whether or not you obtained informed written or verbal consent. If consent was waived for your study, please include this information in your statement as well. 

Reviewers' comments:

Reviewer's Responses to Questions

**Comments to the Author**

1. Is the manuscript technically sound, and do the data support the conclusions?

Reviewer #1: Partly

Reviewer #2: Partly

2. Has the statistical analysis been performed appropriately and rigorously? 

Reviewer #1: Yes

Reviewer #2: Yes

3. Have the authors made all data underlying the findings in their manuscript fully available?

Reviewer #1: Yes

Reviewer #2: Yes

4. Is the manuscript presented in an intelligible fashion and written in standard English?

Reviewer #1: Yes

Reviewer #2: Yes

5. Review Comments to the Author

Reviewer #1: The study entitled” Gender-based heterogeneity of FAHFAs in trained runner “investigated the relationship between baseline circulating FAHFAs in individuals who were self-reported runners relative to various body composition measurements.

The interpretation of the results was often confusing and needs to be rewritten. A few specific examples can be found below.

It is uncertain why “trained runners” is used in the title. How being aerobically trained may have influenced the results was not mentioned in the results or discussion. In addition, from the data presented, training appeared to have no effects on the study outcomes.

The investigators provide no evidence that their subjects were in fact aerobically trained. Participants were selected based solely on self-reported data about the number of times each week they participated in aerobic exercise; three times per week was the minimum inclusion criteria. In addition, the investigators said that participants were able to run for 90 minutes, but no information on this was provided. Without this information or any quantifiable fitness data, it is difficult to accurately call the participants trained or make any inference for how training affected results. (e.g., line 68, page 3).

In addition, the investigators did not include aged-mated non trained participants. Therefore, it is suggested that all mention of training be excluded from this manuscript.

If in fact, these participants were regularly running, no information was provided for the time frame between the last bout of exercise and blood draw. Aerobic exercise is known to modulate circulating fatty acids. In fact, the investigators previously showed that FAHFAs were dynamically regulated during acute exercise in lean participants (line 26, page 1). Without a standardized time frame for this, the acute effect of the last bout of exercise may have influenced the circulating FAHFA values and therefore biased the results.

No information was provided about the current diet of the participants. As with the acute effect of exercise, this could have also influenced circulating FAHFA values and therefore biased the results.

The investigators state that the clustering of FAHFA distributions by BMI support the hypothesis that excess adiposity influences regulation of FAHFA turnover. (lines 93-94, page 4). However, it has been shown that BMI is not a good measure of adiposity as some individuals who present with high values do so because of large LBM. Please consider revising.

Was FAHFAs different when pooling all females versus males studied? Why did the investigators choose to only examine this relationship between NWT males and females? The investigators state that females had a higher FLR than men; infer that this anthropometric index could be driving a difference in FAHFA; but do not show how this affected FAHFAs other than matching for BMI (see previous comment on BMI). They point to the fact that when lean and overweight groups were combined, FAHFA concentrations show the expected negative association with body fay (Figure 2). However, this data was only examined in female subjects. Going back and forth between data sets only confuses the interpretation of the results.

The investigators comment on the fact that visceral adipose tissue is associated with cardiometabolic disease risk (line 47, page 2) and conclude that pathogenic adipose expansion, particularly in VAT, negatively influences circulating FAHFA concentrations. However, the investigators practically ignore to mention in their discussion (and direct analysis) the group (OWT males) who clearly had the most VAT. Information from this group needs to be added to the manuscript and the interpretation of results explained in the discussion.

The sentence on lines 75-765 (page 3) is somewhat confusing and needs further explanation.

The number of subjects was quite low in each group and limits the interpretation of results.

Reviewer #2: This original research manuscript entitled “Gender-based heterogeneity of FAHFAs in trained runners” provides interesting findings regarding serum FAHFA concentrations in relation to gender-based, and anthropometric-based differences in recreationally trained runners. The author group has previously published within this context (i.e., examining the acute exercise response of serum FAHFA in similar groups), and thus have experience within the niche of FAHFA serum concentrations, specifically as they relate to exercise status/capacity and anthropometrics. This article does succeed in its stated goal of determining potential differences in serum FAHFA content between gender, as well as between obese/overweight vs normal weight recreationally trained runners.

Based upon previously published work, as well as indicated by the authors, certain expected differences were observed herein. For example, results from the MS analyses revealed a strong negative correlation in much of the 25 FAHFA species in relation to total fat mass. Interestingly, the authors also pointed out that 5 certain FAHFA species were not correlated with fat mass, but were however correlated with serum glucose, insulin, and HOMA-IR values. There were some findings that the authors indicated to be unexpected; the observed negative relationship between LBM in 14 out of the 25 FAHFA species in NWT vs OWT (males & females combined) appears counter to the paradigm that leanness is positively associated with FAHFA content.

Regarding the LBM-FAHFA negative relationship, it is important to consider that obese individuals often have greater LBM compared to normal-weight counterparts. This stands to reason, as adiposity—and body mass—is accumulated specifically by a net-positive energy balance. The calorie surplus that results in an individual becoming overweight or obese is not exclusive to adiposity, rather increased skeletal muscle is also well-described. This highlights an inherent shortcoming of solely relying upon correlational data to guide future work and/or hypotheses. The authors should likely highlight that a potential explanation of this observed negative relationship between LBM and FAHFA content is that obese individuals often (and within this study itself) exhibit greater total LBM to normal-weight counterparts as a side-effect/consequence of the net positive energy balance responsible for adiposity/obesity. Overfeeding—absent of other direct stimulus (e.g., hypertrophy/resistance training)—can be a potent stimulus for mTOR activity and skeletal muscle growth. Conversely, long-term endurance training expends a significant energy and often results in a net negative energy balance and body weight loss overtime—this includes the loss of LBM. Accordingly, it’s evident that NWT exhibit reduced LBM compared their OWT counterparts. The present manuscript exhibits correlational analyses to examine specific fat mass depots, fat mass, and lean mass, independently. However, there does not appear to be any direct correlational and/or other analyses examining body composition specifically.

Specific Comments:

-The authors should explicitly perform analyses to examine body fat % compared to total mass and total LBM % in relation to FAHFA content. This will better elucidate if the observed negative LBM-to-FAHFA relationship is driven by LBM content alone, OR rather a relationship driven by obesity itself (as obesity itself is associated with > LBM).

-Recent literature highlights some inherent issues associated with the type of MS approach utilized to quantify serum FAHFA. One such paper (Nelson et al., Artifactual FA dimers mimic FAHFA signals in untargeted metabolomics pipelines. J Lipid Res. 2022 May) highlighting FA dimers and artifacts in this type of metabolomics pipeline, specifically cites a previous study conducted by the current authors. However, the authors make no explicit mention of this within the manuscript as either an acknowledgement of the analytical issues inherent in the type of untargeted metabolomics pipelines used in the current manuscript.

-The rigor and technical soundness of this manuscript would be greatly improved with the inclusion of additional/complimentary techniques/analyses. Specifically, this manuscript is almost entirely grounded in one technique (MS shotgun metabolomics), and almost entirely within the analytical context of Pearson correlations. It’s very difficult to draw any solid conclusions from omics and correlations alone. It would be my recommendation to include some form of lipid species examination in addition to the MS data, as a means of confirmation and/or as an adjunct to the present study.

6. PLOS authors have the option to publish the peer review history of their article (what does this mean?). If published, this will include your full peer review and any attached files.

Reviewer #1: No

Reviewer #2: No

---

## [Author Response · Author response to Decision Letter 0]

6 Dec 2023

Point-by-point responses have been attached

---

## [Decision Letter · Decision Letter 1]

21 Feb 2024

Gender-based heterogeneity of FAHFAs in trained runners

PONE-D-23-18993R1

Dear Dr. Crawford,

We’re pleased to inform you that your manuscript has been judged scientifically suitable for publication and will be formally accepted for publication once it meets all outstanding technical requirements.

Kind regards,

Stephen E Alway, Ph.D.

Academic Editor

PLOS ONE

Reviewers' comments:

Reviewer's Responses to Questions

**Comments to the Author**

1. If the authors have adequately addressed your comments raised in a previous round of review and you feel that this manuscript is now acceptable for publication, you may indicate that here to bypass the “Comments to the Author” section, enter your conflict of interest statement in the “Confidential to Editor” section, and submit your "Accept" recommendation.

Reviewer #1: All comments have been addressed

Reviewer #2: All comments have been addressed

2. Is the manuscript technically sound, and do the data support the conclusions?

Reviewer #1: Yes

Reviewer #2: Yes

3. Has the statistical analysis been performed appropriately and rigorously? 

Reviewer #1: I Don't Know

Reviewer #2: Yes

4. Have the authors made all data underlying the findings in their manuscript fully available?

Reviewer #1: Yes

Reviewer #2: Yes

5. Is the manuscript presented in an intelligible fashion and written in standard English?

Reviewer #1: Yes

Reviewer #2: Yes

6. Review Comments to the Author

Reviewer #1: Thank you for the thorough revision of your manuscript and for answering all of my questions and concerns.

Reviewer #2: In response to my original comments, the authors have adequately addressed my concerns in this re-submission. Any additions recommended that were not feasible were acknowledged by the authors and sound reasoning was provided.

7. PLOS authors have the option to publish the peer review history of their article (what does this mean?). If published, this will include your full peer review and any attached files.

Reviewer #1: No

Reviewer #2: No
